# Predicting Forage Quality of Warm-Season Legumes by Near Infrared Spectroscopy Coupled with Machine Learning Techniques

**DOI:** 10.3390/s20030867

**Published:** 2020-02-06

**Authors:** Gurjinder S. Baath, Harpinder K. Baath, Prasanna H. Gowda, Johnson P. Thomas, Brian K. Northup, Srinivas C. Rao, Hardeep Singh

**Affiliations:** 1Department of Plant and Soil Sciences, Oklahoma State University, 371 Agricultural Hall, Stillwater, OK 74078, USA; hardeep.singh@okstate.edu; 2Department of Computer Science, Oklahoma State University, 219 MSCS, Stillwater, OK 74078, USA; hbaath@okstate.edu (H.K.B.); jpthomas@okstate.edu (J.P.T.); 3USDA-ARS, Southeast Area Branch, 114 Experiment Station Road, Stoneville, MS 38776, USA; prasanna.gowda@usda.gov; 4USDA-ARS, Grazinglands Research Laboratory, 7207 W. Cheyenne St., El Reno, OK 73036, USA; brian.northup@usda.gov (B.K.N.); srinivas.rao@usda.gov (S.C.R.)

**Keywords:** partial least square, support vector machine, Gaussian processes, soybean, pigeon pea, guar, tepary bean

## Abstract

Warm-season legumes have been receiving increased attention as forage resources in the southern United States and other countries. However, the near infrared spectroscopy (NIRS) technique has not been widely explored for predicting the forage quality of many of these legumes. The objective of this research was to assess the performance of NIRS in predicting the forage quality parameters of five warm-season legumes—guar (*Cyamopsis tetragonoloba*), tepary bean (*Phaseolus acutifolius*), pigeon pea (*Cajanus cajan*), soybean (*Glycine max*), and mothbean (*Vigna aconitifolia*)—using three machine learning techniques: partial least square (PLS), support vector machine (SVM), and Gaussian processes (GP). Additionally, the efficacy of global models in predicting forage quality was investigated. A set of 70 forage samples was used to develop species-based models for concentrations of crude protein (CP), acid detergent fiber (ADF), neutral detergent fiber (NDF), and in vitro true digestibility (IVTD) of guar and tepary bean forages, and CP and IVTD in pigeon pea and soybean. All species-based models were tested through 10-fold cross-validations, followed by external validations using 20 samples of each species. The global models for CP and IVTD of warm-season legumes were developed using a set of 150 random samples, including 30 samples for each of the five species. The global models were tested through 10-fold cross-validation, and external validation using five individual sets of 20 samples each for different legume species. Among techniques, PLS consistently performed best at calibrating (R^2^*_c_* = 0.94–0.98) all forage quality parameters in both species-based and global models. The SVM provided the most accurate predictions for guar and soybean crops, and global models, and both SVM and PLS performed better for tepary bean and pigeon pea forages. The global modeling approach that developed a single model for all five crops yielded sufficient accuracy (R^2^*_cv_*/R^2^*_v_* = 0.92–0.99) in predicting CP of the different legumes. However, the accuracy of predictions of in vitro true digestibility (IVTD) for the different legumes was variable (R^2^*_cv_*/R^2^*_v_* = 0.42–0.98). Machine learning algorithms like SVM could help develop robust NIRS-based models for predicting forage quality with a relatively small number of samples, and thus needs further attention in different NIRS based applications.

## 1. Introduction

Perennial warm-season grasses, such as bermudagrass (*Cynodon dactylon*), old world bluestems (*Bothriochloa* spp.), and bahiagrass (*Paspalum notatum*), serve as major summer forage resources for grazing stocker cattle in the southern United States (US). While capable of producing large amounts of biomass, these perennial grasses often show a decline in forage quality with their maturation towards the mid-late summer growing season and do not meet the nutritional needs of grazing stocker cattle for the entire season [1,2]. Legumes, being high-quality forages, can be adopted to offset the summer slump in forage quality, and enhance the efficiency of forage-based beef production systems. Further, the continued increase in the cost of nitrogen fertilizers has added to the interest of producers in utilizing legume crops as forage in many regions across the US. In response, extensive research in the southern US over the last decade has focused on evaluating warm-season annual legumes as summer forage resources that can be grown in rotation with winter-wheat (*Triticum aestivum* L.) [3,4,5,6]. In more recent years, several legumes have received increased attention due to their capabilities of generating high amounts of biomass under the limited moisture conditions that prevail in the southern US [7].

Quantifying the quality of forage in pastures is crucial for both agriculture research and forage management, including cattle grazing and harvesting. However, the determination of the different parameters of forage quality, such as crude protein (CP), neutral detergent fiber (NDF), acid detergent fiber (ADF), and in vitro true digestibility (IVTD), by classical analytical techniques is time-consuming and expensive, especially when numerous samples are required. The vast evolution of computers and multivariate statistical techniques has enabled the use of near infrared spectroscopy (NIRS) in assessing the quality parameters of many forages. The NIRS method is quick, inexpensive, and facilitates timely decision-making related to grazing periods. The technique is based on interactions between light reflectance in the wavelength ranging between 750–2500 nm and organic compounds in the plant biomass [8]. The method of applying NIRS to predict forage quality involves analyzing a particular forage with both traditional lab analysis and NIRS, and then developing a predictive equation by pairing the information in a calibration dataset (Figure 1). The NIRS has been widely used in forage quality predictions of crops including alfalfa (*Medicago sativa*) [9], maize (*Zea mays*) [10], ryegrass (*Lolium multiflorum*) [11], tall fescue (*Festuca arundinacea*) [12], and other species. However, the technique has been underutilized to provide predictions of forage quality for many warm-season legumes.

As developed for other forage crops, well-calibrated NIRS species-based models for warm-season legumes could be useful tools to quickly asses the forage quality of different legume species grown under a range of environmental or management settings, or harvested at different stages of growth, or cutting or grazing height. Therefore, it is necessary to examine the effectiveness of NIRS in predicting forage characteristics of some important warm-season legumes. This work includes species such as guar (*Cyamopsis tetragonoloba*), tepary bean (*Phaseolus acutifolius*), soybean (*Glycine max*), and pigeon pea (*Cajanus cajan*), given past research, and the potential for expansion of use of these species across the southern US and other similar environments. There are also other warm-season legumes that may be capable of providing high-quality forage for summer grazing [3,13,14]. However, developing NIR calibration equations for every species can become challenging for public or private laboratories that test forage quality. Generally, accurate chemical analyses of a large number of samples is not readily available or feasible to develop calibrations, especially when novel legume species are involved. In response to challenges related to developing species-based calibrations, global models developed from samples of ranges of different warm-season legumes can prove useful, if such calibrations provide sufficiently accurate predictions.

Several calibration techniques are known to perform well in the application of NIRS in estimating forage quality and are generally available in most chemometric packages [15]. Partial least squares (PLS) is among the most commonly used methods, where least square algorithms are used to compute regressions [16]. In contrast, a comparatively novel and robust machine learning algorithm, support vector machine (SVM), has been gaining attention for NIRS calibrations [15]. Further, the Gaussian processes (GP) have provided better calibration results than PLS and SVM, in some cases [17,18]. However, tests of these calibration techniques on wide ranges of common and more novel legumes are required to define their function.

The combination of NIRS and machine learning calibration techniques could serve as an effective tool to streamline the monitoring efforts in warm-season legumes by eliminating the need for classical forage analytical methods. Therefore, the objective of this research was to evaluate the performance of NIRS in predicting the forage quality of four warm-season legumes (guar, tepary bean, pigeon pea, and soybean), using three different calibration techniques—PLS, SVM, and GP—on individual species bases. Additionally, the efficacy of global calibrations of these techniques, developed by combining datasets of all four species and mothbean (*Vigna aconitifolia*), was tested using different independent datasets of five species.

## 2. Materials and Methods

### 2.1. Materials

Forage samples used in the study (*n* = 410) were collected as parts of two different field experiments conducted at the USDA-ARS Grazinglands Research Laboratory near El Reno, Oklahoma, US (35.57° N, 98.03° W, elevation 414 m). Ninety samples each for guar and tepary bean, and 50 mothbean samples were collected from field experiments conducted during the summers of 2017 and 2018. An additional 90 samples of both soybean and pigeon pea were obtained from two long-term experiments (2001–2008) conducted in the same location [3,19]. In all three experiments, aboveground biomass was collected from randomly clipped 0.5 m row lengths from experimental plots at 15-day intervals, starting at 45 days after planting. Apart from whole plant samples, a major proportion of collected biomass samples in these experiments were separated into leaf, stem, and pods fractions before laboratory analysis.

### 2.2. Laboratory and NIRS Analysis

All leaf, stem, pod, and whole plant samples were oven-dried at 60 °C until a constant weight. Dry samples were ground to pass a 2-mm filter using a Wiley grinding mill. Total nitrogen concentration in each sample was determined by the flash combustion method (Model Vario Macro, Elementar Americas, Inc., Mt. Laurel, NJ, USA) and then converted into CP by multiplying with a factor of 6.25 (Table 1). The IVTD was obtained for each sample by following the Daisy Digester procedures (ANKOM Technology, Macedon, NY, USA). The NDF and ADF concentrations were only determined in samples of guar and tepary bean, in accordance with the batch fiber analyzer techniques (ANKOM Technology, Macedon, NY, USA).

Aliquots of ground samples were filled into ring cups to eliminate voids. Spectral reflectance (R) of monochromatic light, averaged over 10 spectra per sample, were collected by scanning spectrophotometer (Model SpectraStar 2600 XT-R, Unity Scientific, Columbia, MD, USA). Spectral data were obtained as the logarithm of the inverse of reflectance [log(1/R)] at 1-nm interval over the range of 680–2600 nm.

### 2.3. Calibration Techniques

**Partial least squares (PLS)** is an extensively used class of statistical methods, which includes regression, classification, and dimension reduction techniques. It uses latent variables, also called score vectors, to model the relationship between input and response variables. In the case of regression problems, PLS first generates the latent variables from the given data and uses them as new predictor variables. There are different types of PLS, based on techniques employed to extract the latent variables. Two approaches are used to extend PLS for modeling non-linear relations among data. The first approach is to reformulate the linear relationship between score vectors, u and v, by a non-linear model:(1)v=g(u)+h=g(X,w)+h
where g is the continuous function that models the existing non-linear relation. Generally, g is modeled using artificial neural networks, smoothing splines, polynomial, or radial basis functions. Remaining variables h and w denote a residual vector and a weight vector, respectively.

The second PLS approach is to apply kernel-based learning. The kernel PLS method transforms the input space data to higher dimensional feature space and linearly estimates PLS in that space. To avoid the mapping function Φ from projecting data to feature space, PLS applies the kernel trick which uses the fact that a value of the inner product of two vectors x and y in feature space can be calculated using a kernel function k(x,y) [20]:(2)$$k\left(x,y\right)=\Phi {\left(x\right)}^{T}\Phi \left(y\right)$$

By using the kernel function, score vectors (u and v) can be identified and used to define the non-linear relationship. The kernel PLS approach is used to model complex non-linear relations easily in terms of implementation and computation.

**Gaussian processes (GP)** are kernel-based, probabilistic, non-parametric regression models. A Gaussian process involves a set of random variables such that every finite number of those variables possess joint Gaussian distributions. A Gaussian process, f(x), can be described using a mean function m(x) and a covariance function $k\left(x,x^{\prime }\right).$ The covariance function defines the smoothness of responses, and the basis function Φ projects the input space vector x to a higher dimension feature space vector Φ(x). A Gaussian process regression (GPR) model describes the response by using latent variables from a Gaussian process. A GPR model is represented as:(3)$$\Phi {\left(x_{i}\right)}^{T}w+f\left(x\right)$$
where $f\left(x\right)\;\sim \;GP\left(0,k\left(x,x^{\prime }\right)\right)$, and f(x) are from a zero mean GP having a covariance function, $k\left(x,x^{\prime }\right)$ [21]. The covariance is specified by kernel parameters, which are also known as hyperparameters. GPR is a probabilistic model, and an instance of response y is:(4)$$P\left(y_{i}|f\left(x_{i}\right),x_{i}\right)\;\sim \;N\left(y_{i}|\Phi {\left(x_{i}\right)}^{T}w+f(x_{i}\right),\sigma^{2})$$

GPR is non-parametric as there is a latent variable $f\left(x_{i}\right)$ for each observation $x_{i}$. Noise variance $\sigma^{2}$, basis function coefficients w, and hyperparameters of the kernel can be estimated from the data while training the GPR model.

**Support vector machine (SVM)** is a popular machine-learning algorithm used for identifying linear as well as non-linear dependency between input vectors and outputs. SVMs are non-parametric models, which means parameters are selected, estimated, and tuned in such a way that the model capacity matches the data complexity [21]. Generally, SVM starts by observing the multivariate inputs X and outputs Y, estimates its parameters w, and then learns the performed mapping function y=f(x,w), which approximates the underlying dependency between inputs and responses. The obtained function, also known as a hyperplane, must have a maximal margin (for classification) or the error of approximation (for regression) to predict the new data. In the case of SVM regression, Vapnik’s error (loss) function is used with ε-insensitivity. It finds a regression function f(x) that deviates from the actual responses (y) by values no more than ε and is considerably flat at the same time.

For non-linear regression problems, SVM maps the input space to feature space (a higher dimension space) using a mapping Φ(x) to find a linear regression hyperplane in that space. However, there is no need to know the mapping Φ, as the kernel function $k\left(x_{i},x_{j}\right),$ which is the inner product of the vectors $\Phi \left(x_{i}\right)$ and $\Phi \left(x_{j}\right)$, can be used to find the optimal regression hyperplane in extended space. There are many kernel functions available to describe non-linear regressions, such as the polynomial kernel, RBF kernel, Gaussian Kernel, normalized polynomial kernel, etc. The learning problem in classification as well as in regression, leads to solving the quadratic programming (QP). The sequential minimal optimization (SMO) is considered as the most popular optimizer for solving SVM problems [22]. It divides the large QP problem into a set of small QP problems and analytically solves them.

### 2.4. Performance Evaluation

Apart from calibration, 10-fold cross-validations and external validations were conducted to assess the performance of the calibration techniques. The 10-fold cross-validation is a unique statistical way of performance evaluations of machine learning models in which ten repeated hold-out executions are obtained and averaged. In each execution, the model is trained with 90% of the data points and tested with the remaining 10%, and thus every data point is taken nine times for training and once for testing the model. For each species-based model, the original dataset of 90 samples for each species was split into two subsets (Figure 2). A subset of 70 samples was used for running calibration and 10-fold cross-validation. The other subset of 20 remaining samples was used only for external validation and neither used in calibration nor cross-validation of any model. For the global model, the original dataset consisted of 250 samples, involving 50 samples each of guar, tepary bean, soybean, pigeon pea, and mothbean. These samples were divided into six subsets (Figure 2). One random subset of 150 samples (30 samples per species) was employed for calibration as well as 10-fold cross-validation. Each of the remaining five subsets, comprising 20 samples of individual species, was used for external validation.

Coefficients of determination, being upper-bounded by 1.0, are often adopted for meaningful comparisons across different models and therefore was used here as an estimate of prediction accuracy. To be precise, coefficient of determination in calibration (R^2^*_c_*), coefficient of determination in cross-validation (R^2^*_cv_*), and coefficient of determination in validation (R^2^*_v_*) were used for direct computation of the variance in the data captured at calibration, cross-validation, and external validation, respectively by each model. Additionally, root mean squared error estimation was also presented for comparing models, which were termed as RMSE*_c_*, RMSE*_cv_*, and RMSE*_v_* for calibration, cross-validation, and external validation, respectively.

### 2.5. Software

Regression models were calibrated, cross-validated, and externally validated using the *Weka* software, version 3.8 [23]. Weka is a suite of machine learning algorithms and is widely used for data mining. For implementing PLS, we used the PLS classifier package in Weka, which uses the prediction capabilities of PLSFilter. The PLSFilter runs the PLS regression on the given set of data and computes the beta matrix for prediction. By default, missing values are replaced, and the data are centered. For GP implementation, the Gaussian classifier for regression without hyperparameter-tuning was used. The kernel for the Gaussian classifier was configured as a polynomial. By default, missing values were replaced by the global mean. The SMOReg classifier was used to implement SVM in Weka. The classifier used the polynomial kernel and RegSMOImproved optimizer to learn SVM for regression. All remaining parameters, such as batch size, debugging, and filter type, which do not check capabilities, noise, etc., were kept as default.

## 3. Results and Discussion

The prediction accuracy of calibrated models is discussed by comparing their cross-validation (R^2^*_cv_*) and external validation (R^2^*_v_*) results to a scale proposed for NIRS calibrations [24]. According to the scale, the performance of a model is considered excellent if the R^2^ of validations is greater than 0.95, and the resultant model can be used in any application. A model is assumed satisfactory with R^2^ ranging from 0.9–0.95 and would be usable for most applications involving quality assurance. Models with R^2^ ranging between 0.8–0.9 are considered moderately successful and can be used with caution for most applications, including research.

### 3.1. Guar

The chemical analysis of guar samples showed wide variability in parameters that define forage quality for different components (leaf, stem, or pod) of plant sampled at different growth stages (Table 1). The CP content for all 90 (70 + 20) guar samples ranged from 3.7% to 34.9%, while NDF concentrations ranged from 16.8% to 75.8%, ADF concentrations ranged from 8.9% to 62.9%, and IVTD from 40.3% to 95.2%.

Among the three techniques, the PLS technique performed best at calibrating each of the four forage quality parameters in guar with R^2^*_c_* of 0.98–0.99, though calibration results of SVM (R^2^*_c_* = 0.94–0.98) were also comparable (Table 2). While GP had a comparatively lower calibration accuracy with R^2^*_c_* ranging between 0.88–0.91 for IVTD, NDF and ADF, and R^2^*_c_* of 0.95 for CP of guar samples. Although PLS provided best calibrations out of the three, SVM gave better prediction accuracy in both cross-validation and external validation of all four indices of forage quality for guar. Thee GP approach generated the lowest R^2^*_cv_* for all four parameters and R^2^*_v_* for NDF and ADF.

Among forage quality parameters, the greatest prediction accuracy was recorded for CP by all three techniques with R^2^*_cv_* of 0.93–0.97 and R^2^*_v_* of 0.93–0.98 (Table 2). In comparison, only the SVM technique resulted in a satisfactory prediction accuracy (R^2^*_cv_* = 0.92; R^2^*_v_* = 0.94) for NDF, based on the proposed scale [24]. Both the SVM and PLS techniques showed excellent accuracy at predicting ADF with R^2^*_cv_* and R^2^*_v_* between 0.94–0.96, while GP produced R^2^*_cv_* of 0.86. All three techniques resulted in relatively low prediction accuracy for IVTD, with R^2^*_cv_* ranging from 0.81–0.83. Overall, performances of SVM was most satisfactory among the three calibration methods, and it can be employed in NIRS-based prediction of CP, ADF, and NDF of guar. In contrast, use of IVTD predictions of guar would require caution, based on the type of application.

While currently a minor crop in the southern US, guar has a proven potential to serve as a multi-purpose legume and has potential for expansion in use. Guar is a common crop in regions of the Indian subcontinent, Africa, North and South America, and Australia [25]. Guar has been gaining attention as a forage resource in the southern US due to its capability of producing high N biomass under limited water conditions [3,5]. Therefore, this first report investigating the application of NIRS in guar would encourage the utilization of the technique in its research and forage management.

### 3.2. Tepary Bean

Results from the laboratory analysis of tepary bean samples showed high variability in all four of the quality indices, though the observed ranges were narrower than guar (Table 1). The concentration of CP varied from 4.5–31.1%, while NDF ranged from 22.9% to 71.6%. In contrast, ADF and IVTD ranged between 15.3–59.2% and 55.9–93.2%, respectively. Best calibration results for tepary bean were recorded using the PLS technique, with R^2^*_c_* of 0.98–0.99 (Table 3). Whereas, neither SVM nor PLS clearly resulted in better predictions for all quality indices when cross-validated and externally validated.

All calibration techniques showed best results at predicting CP in tepary bean samples with a R^2^*_cv_* or R^2^*_v_* above 0.90 among the forage quality characteristics (Table 3). The SVM technique resulted in the lowest RMSE*_cv_* value (1.74) for cross-validation of CP, whereas PLS had the lowest RMSE*_v_* of 1.35 for external validation among the three techniques. In contrast, PLS showed the lowest RMSE*_cv_* values of 5.09 and 3.97 and SVM had the lowest RMSE*_v_* of 4.03 and 2.23 for NDF and ADF, respectively. Both PLS and SVM produced satisfactory results at predicting ADF concentration in tepary bean with R^2^*_cv_* of 0.86–0.89 and R^2^*_v_* of 0.92–0.95 compared to GP, while all three techniques had comparatively low performance at predicting NDF in tepary bean with R^2^*_cv_* and R^2^*_v_* of 0.72–0.84 and 0.75–0.84, respectively.

In comparison to ADF, the NDF concentration of tepary samples were less accurately predicted by all three techniques (Table 3). Similar differences between prediction accuracy of ADF and NDF were also noticed for guar in this study, and also reported earlier in NIRS studies involving *Brassica napus* [26], *Lolium multiflorum* [11], and *Oryza sativa* [27]. Though PLS performed better at predicting IVTD in tepary bean compared to other two, all three techniques resulted in relatively low prediction accuracy with R^2^_c*v*_ and R^2^*_v_* ranging between 0.75–0.79 and 0.75–0.88, respectively. Overall, both PLS and SVM could be considered as good among three tested techniques and hence can be employed for satisfactory predictions of CP and ADF in tepary bean. While prediction results of NDF and IVTD would need some caution if calibrations are developed with similar sample sizes (*n* = 70) as used in this study.

Tepary bean is a vining, warm-season legume species originated from the areas of the southwestern United States and northwestern Mexico, that may have value for multiple uses in dryland agricultural systems. Due to its spreading growth habit, and the ability to generate high N biomass with limited soil moisture, tepary bean could be an ideal summer forage for the Southern Great Plains [14]. This first study investigating the application of NIRS to attributes of forage quality in tepary bean showed that the technique could aid in quantifying its role in meeting animal nutrition needs.

### 3.3. Soybean

All three techniques (PLS, SVM, and GP) gave excellent accuracies at calibrating CP and IVTD of soybean samples with PLS again performing the best out of three with a R^2^*_c_* greater than 0.98 (Table 4). Among three techniques, SVM performed best at predicting CP with RMSE*_cv_* and RMSE*_v_* of 1.85 and 1.78, respectively, followed closely by PLS. All three calibration techniques produced better predictions of IVTD in soybean (R^2^*_cv_* > 0.84 and R^2^*_v_* > 0.89), compared to prediction accuracies obtained for guar and tepary bean. As observed for CP, SVM performed better than the other techniques in cross-validation (R^2^*_cv_* = 0.89) of IVTD, while the other two techniques performed better in external validation (R^2^*_v_* of 0.92–0.93). All three techniques can be employed for rapid NIR-based predictions of CP and IVTD in soybean forage samples, with SVM would be the best choice.

Soybean was initially introduced as a forage into the US in the 19th Century, but is now one of the most widely grown grain legumes in the Southern Great Plains [14]. In the last two decades, there has been increased interest from researchers in utilizing soybean as a summer forage in the US [28,29,30]. Hence the need for rapid and low-cost techniques for estimating forage quality. The NIRS technique has not been exploited for forage quality predictions in soybean. A single report investigated modified PLS and multiple scatter correction methods for NIR predictions of CP, NDF, and ADF concentrations, using 353 soybean samples collected at one (R6) growth stage [31]. In comparison, calibrations developed in the present study, used data on IVTD and CP with just 70 soybean samples collected across a range of different growth stages. Thus, our observed ranges for CP (4.1–39.7) and IVTD (42.4–99.3%) were more diverse (Table 1). The accuracies (R^2^*_cv_* or R^2^*_v_* > 0.92) obtained in predicting CP in soybean forage by all three techniques were higher than the values reported [31], despite large differences in sample sizes (N = 70 vs. 353) used for developing calibrations. Therefore, this study showed machine learning algorithms could develop robust NIRS calibrations for precise analysis of forage quality of soybean with small sample sizes.

### 3.4. Pigeon Pea

Laboratory analyses for the current study showed wide variability in both CP (4.5–32.5%) and IVTD (30.7–91.1%) for forage samples of pigeon pea (Table 1). The CP concentration in pigeon pea was accurately calibrated (R^2^_*c*_ > 0.95) by each of the three techniques (Table 5). All three techniques resulted in CP predictions with R^2^*_cv_* and R^2^*_v_* greater than 0.96. Both PLS and SVM also showed greater accuracies in predicting IVTD of pigeon pea with R^2^*_cv_* and R^2^*_v_* ranges of 0.91–0.92 and 0.96–0.97, respectively. Although lower than PLS and SVM, the performance (R^2^*_cv_* = 0.86) of GP-based calibrations were moderately satisfactory in IVTD predictions, following the proposed scale [24]. Overall, both PLS and SVM would provide excellent options for NIR predictions of CP and IVTD in pigeon pea.

Pigeon pea is another legume species that has seen the development of a range of cultivars for different uses in its home range, and areas of greater cultivation. This includes research on the value of cultivars of pigeon pea in the US for forage, grain, and pasture productivity [4,32]. Pigeon pea has a high degree of heat and drought tolerance, and the capacity for high levels of forage production in the US and other tropical and sub-tropical regions.

While pigeon pea is a broadly grown crop in much of the world, there was only one preliminary report that discussed the possible use of NIRS techniques to predict forage quality of pigeon pea [33]. That report used limited numbers of samples (*n* = 48), involving leaves and branches, that were mostly collected at one growth stage for calibrations of CP, NDF, and ADF concentrations; however, no validations were performed [33]. In contrast, the present study undertook both calibrations and validations using 90 (70 + 20) pigeon pea samples, involving leaves, stems, or seed pods, collected at different growth stages during a long-term experiment. Further, we investigated the NIR-based predictions of IVTD, which is assumed as an important quality parameter in pigeon pea forage [4]. Therefore, this study confirms that NIRS techniques could be effective tools for predicting forage quality of pigeon pea.

### 3.5. Global Calibrations

Global calibrations for CP and IVTD of warm-season legumes were developed with 150 samples, which included 30 samples each of guar, tepary bean, soybean, pigeon pea, and mothbean (Figure 2). As observed with the species-based calibrations for the four different legumes, the PLS technique performed best out of the three techniques for global calibrations both CP and IVTD (R^2^*_c_* of 0.97 and 0.94, respectively), while the GP technique was the least accurate (Table 6). In comparison, cross-validation of global models showed the SVM approach provided the greatest prediction accuracy for both CP (R^2^*_cv_* = 0.94) and IVTD (R^2^*_cv_* = 0.86), followed closely by PLS. Therefore, based on cross-validation results, the performance of global calibrations developed using SVM and PLS were satisfactory at predicting CP, and moderately satisfactory for IVTD.

When global calibrated models were validated using different external datasets for each of the five legume species, the predictions for CP by all three techniques resulted in sufficient accuracies with R^2^*_v_* ranging between 0.91–0.97 (Table 6). The SVM technique showed higher accuracy compared to the others in predicting CP, with the exception of guar, where the PLS approach provided slight improvements. Among species, the best CP predictions were noted for pigeon pea (R^2^*_v_* values of 0.98–0.99) for all three techniques. In contrast, IVTD predictions were not consistently accurate across all five species. The greatest accuracy was observed for IVTD predictions in pigeon pea with R^2^*_v_* of 0.97–0.98 under SVM and PLS. The lowest accuracy in predicting IVTD was noted for mothbean (R^2^*_v_* between 0.65–0.69 by SVM and PLS; 0.42 by GP). The best prediction accuracies for IVTD of soybean (R^2^*_v_* = 0.82–0.86 for all three techniques) and guar (PLS; R^2^*_v_* = 0.81) were moderately satisfactory. However, the performance of all three techniques was satisfactory at predicting IVTD in tepary bean (R^2^*_v_* of 0.91–92), which was better than the specific models developed for tepary bean (Table 3).

Overall, global-calibrated models for CP have the potential to offer sufficient prediction accuracies that are comparable to species-based calibration models. Diverse calibration sets that contain different legume species may allow the creation of robust, generalized models that provide predictions similar to species-based models. In some cases, global models may be capable of providing more accurate predictions, as was observed for IVTD predictions of tepary bean in this study.

The application of accurate globally calibrated models would be extremely useful for a broad range of end-users. They would reduce or eliminate the large amounts of time and other resources required to perform chemical analyses or the development and use of separate calibration sets for every species. However, adopting the global calibration approach for IVTD may not provide satisfactory predictions for all species. Some of the issues related to the low level of performance of calibrations for IVTD may be variability associated with using techniques that rely on rumen fluids in laboratory analyses [34]. Therefore, further investigations are required to compare the performance of global calibrations developed for IVTD of warm-season legumes derived using both rumen fluid and cellulose degradation methods.

## 4. Conclusions

The statistics obtained for calibration, cross-validation, and external validation in this study demonstrated that NIRS techniques could be effective for supplying rapid and accurate predictions of most attributes of forage quality (cell wall fractions, crude protein) for different warm-season legumes. Further, the applications of NIRS technique to guar, tepary bean, and mothbean represent the first reports of such tools to provide estimates of forage quality for these species. Though similar to PLS, the SVM technique performed consistently well in predicting quality parameters of five warm-season legumes under both species-based and global calibration strategies. The global calibration approach can be a useful approach for predicting CP in warm-season legumes, and reduce the time and resources required for traditional chemical analysis in the use of separate calibration equations for each species. However, the global model for IVTD was not accurate for all species. Further model development based on other analytical procedures may improve the consistency and reliability of the global approach. Machine learning algorithms like SVM could also allow the development of robust models with a relatively small number of samples. Additional research is required to refine the SVM approach for different NIRS applications.

## Figures and Tables

**Figure 1 sensors-20-00867-f001:**
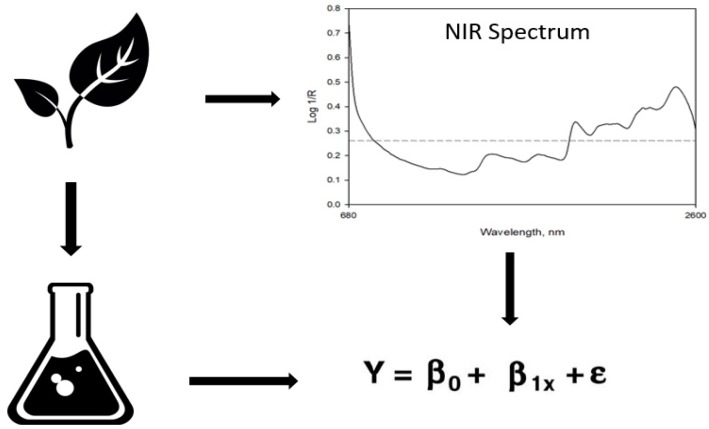
Illustration of the procedure used for applying near infrared spectroscopy (NIRS) technique in forage quality predictions.

**Figure 2 sensors-20-00867-f002:**
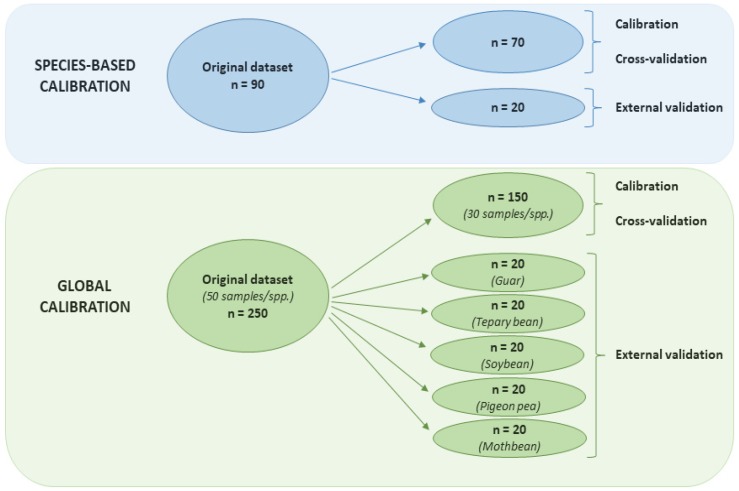
Diagram of the datasets, calibration, and different validation processes used in two calibration strategies.

**Table 1 sensors-20-00867-t001:** Summary statistics of lab datasets used for calibration, cross-validation and external validation of crude protein (CP), neutral detergent fiber (NDF), acid detergent fiber (ADF), and in vitro true digestibility (IVTD) of four warm season legumes.

Species	Parameter	Calibration and Cross-Validation (*n* = 70)				External Validation (*n* = 20)			
		Min	Max	Mean	SD	Min	Max	Mean	SD
		------------------------- (%) -------------------------							
Guar	CP	3.94	34.87	17.66	8.66	3.69	33.56	15.07	9.50
	NDF	16.83	70.80	37.57	16.95	22.95	75.78	45.94	17.82
	ADF	8.90	58.39	27.19	15.29	12.79	62.93	34.70	16.57
	IVTD	40.35	95.22	79.27	14.11	42.96	94.37	73.38	16.08
Tepary bean	CP	4.50	31.12	15.76	7.78	5.94	30.25	19.35	8.13
	NDF	22.90	71.57	48.34	12.31	25.52	60.95	43.90	10.21
	ADF	15.32	59.16	34.92	11.99	17.08	48.16	30.36	10.39
	IVTD	55.88	93.16	75.50	10.83	60.23	92.56	81.34	8.56
Soybean	CP	4.15	39.75	21.16	11.03	6.31	36.12	19.73	8.94
	IVTD	42.45	99.30	78.25	16.28	57.66	98.31	80.21	12.38
Pigeon pea	CP	4.52	32.48	16.30	8.77	6.24	28.64	15.62	7.41
	IVTD	30.71	91.08	61.55	19.28	33.31	82.89	59.76	16.40

*n*, number of samples; Min, minimum value; Max, maximum value; SD, standard deviation.

**Table 2 sensors-20-00867-t002:** Calibration, cross-validation, and external validation statistics obtained for crude protein (CP), neutral detergent fiber (NDF), acid detergent fiber (ADF), and in vitro true digestibility (IVTD) in guar using three calibration techniques.

Parameter	Method	Calibration (*n* = 70)		Cross-Validation (*n* = 70)		External Validation (*n* = 20)	
		R^2^*_c_*	RMSE*_c_*	R^2^*_cv_*	RMSE*_cv_*	R^2^*_v_*	RMSE*_v_*
CP	GP	0.95	1.84	0.93	2.20	0.96	2.12
	PLS	0.99	0.78	0.95	1.97	0.93	2.52
	SVM	0.98	1.23	0.97	1.56	0.98	1.27
NDF	GP	0.90	5.53	0.84	6.73	0.90	6.98
	PLS	0.98	2.17	0.85	6.66	0.93	5.52
	SVM	0.94	3.98	0.91	5.08	0.94	4.67
ADF	GP	0.91	4.79	0.86	5.77	0.92	6.02
	PLS	0.99	1.18	0.95	3.36	0.94	4.23
	SVM	0.97	2.46	0.95	3.51	0.96	3.78
IVTD	GP	0.88	4.92	0.81	6.10	0.93	5.63
	PLS	0.98	2.15	0.81	6.69	0.87	5.66
	SVM	0.94	3.51	0.83	5.88	0.94	4.19

GP, Gaussian processes; PLS; partial least square; SVM, support vector machine; R^2^*_c_*, determination coefficient in calibration; RMSE*_c_*, root mean square error in calibration; R^2^*_cv_*, determination coefficient in cross-validation; RMSE*_cv_*, root mean square error in cross-validation; R^2^*_v_*, determination coefficient in external validation; RMSE*_v_*, root mean square error in external validation.

**Table 3 sensors-20-00867-t003:** Calibration, cross-validation, and external validation statistics obtained for crude protein (CP), neutral detergent fiber (NDF), acid detergent fiber (ADF), and in vitro true digestibility (IVTD) in tepary bean using three calibration techniques.

Parameter	Method	Calibration (*n* = 70)		Cross-Validation (*n* = 70)		External Validation (*n* = 20)	
		R^2^*_c_*	RMSE*_c_*	R^2^*_cv_*	RMSE*_cv_*	R^2^*_v_*	RMSE*_v_*
CP	GP	0.94	1.89	0.90	2.42	0.94	2.20
	PLS	0.99	0.68	0.93	2.03	0.98	1.35
	SVM	0.97	1.35	0.95	1.74	0.94	1.94
NDF	GP	0.85	4.96	0.75	6.22	0.75	5.10
	PLS	0.98	1.64	0.84	5.09	0.75	5.53
	SVM	0.94	2.97	0.72	7.01	0.84	4.03
ADF	GP	0.87	4.60	0.78	5.62	0.86	3.90
	PLS	0.98	1.47	0.89	3.97	0.92	3.34
	SVM	0.96	2.45	0.86	4.52	0.95	2.23
IVTD	GP	0.87	4.02	0.75	5.39	0.75	4.25
	PLS	0.98	1.55	0.79	5.00	0.88	2.89
	SVM	0.93	2.86	0.75	5.70	0.82	3.82

GP, Gaussian processes; PLS; partial least square; SVM, support vector machine; R^2^*_c_*, determination coefficient in calibration; RMSE*_c_*, root mean square error in calibration; R^2^*_cv_*, determination coefficient in cross-validation; RMSE*_cv_*, root mean square error in cross-validation; R^2^*_v_*, determination coefficient in external validation; RMSE*_v_*, root mean square error in external validation.

**Table 4 sensors-20-00867-t004:** Calibration, cross-validation, and external validation statistics obtained for crude protein (CP) and in vitro true digestibility (IVTD) in soybean using three calibration techniques.

Parameter	Method	Calibration (*n* = 70)		Cross-Validation (*n* = 70)		External Validation (*n* = 20)	
		R^2^*_c_*	RMSE*_c_*	R^2^*_cv_*	RMSE*_cv_*	R^2^*_v_*	RMSE*_v_*
CP	GP	0.92	4.63	0.87	5.78	0.92	3.78
	PLS	0.98	2.16	0.84	6.92	0.93	3.46
	SVM	0.94	3.92	0.89	5.28	0.89	4.09
IVTD	GP	0.96	2.14	0.94	2.71	0.92	2.53
	PLS	0.99	0.80	0.96	2.05	0.94	2.24
	SVM	0.99	1.26	0.97	1.85	0.96	1.78

GP, Gaussian processes; PLS; partial least square; SVM, support vector machine; R^2^*_c_*, determination coefficient in calibration; RMSE*_c_*, root mean square error in calibration; R^2^*_cv_*, determination coefficient in cross-validation; RMSE*_cv_*, root mean square error in cross-validation; R^2^*_v_*, determination coefficient in external validation; RMSE*_v_*, root mean square error in external validation.

**Table 5 sensors-20-00867-t005:** Calibration, cross-validation, and external validation statistics obtained for crude protein (CP) and in vitro true digestibility (IVTD) in pigeon pea using three calibration techniques.

Parameter	Method	Calibration (*n* = 70)		Cross-Validation (*n* = 70)		External Validation (*n* = 20)	
		R^2^*_c_*	RMSE*_c_*	R^2^*_cv_*	RMSE*_cv_*	R^2^*_v_*	RMSE*_v_*
CP	GP	0.98	1.37	0.96	1.73	0.96	1.69
	PLS	1.00	0.43	0.97	1.46	0.98	1.02
	SVM	0.99	0.84	0.98	1.17	0.98	1.12
IVTD	GP	0.95	4.51	0.86	7.18	0.97	2.95
	PLS	0.99	1.93	0.92	5.49	0.96	3.09
	SVM	0.97	3.31	0.91	5.86	0.97	2.85

GP, Gaussian processes; PLS; partial least square; SVM, support vector machine; R^2^*_c_*, determination coefficient in calibration; RMSE*_c_*, root mean square error in calibration; R^2^*_cv_*, determination coefficient in cross-validation; RMSE*_cv_*, root mean square error in cross-validation; R^2^*_v_*, determination coefficient in external validation; RMSE*_v_*, root mean square error in external validation.

**Table 6 sensors-20-00867-t006:** Calibration, cross-validation, and external (species) validation statistics of global models obtained for crude protein (CP) and in vitro true digestibility (IVTD) in warm-season legumes using three calibration techniques.

Method		Calibration (*n* = 150)		Cross-Validation (*n* = 150)		External Validation (*n* = 20)									
						Guar		Tepary Bean		Soybean		Pigeon Pea		Mothbean	
		R^2^*_c_*	RMSE*_c_*	R^2^*_cv_*	RMSE*_cv_*	R^2^*_v_*	RMSE*_v_*	R^2^*_v_*	RMSE*_v_*	R^2^*_v_*	RMSE*_v_*	R^2^*_v_*	RMSE*_v_*	R^2^*_v_*	RMSE*_v_*
CP	GP	0.92	2.15	0.89	2.46	0.93	2.42	0.95	2.72	0.91	3.95	0.98	2.21	0.94	3.36
	PLS	0.97	1.15	0.92	2.02	0.94	2.36	0.94	2.49	0.94	2.47	0.98	2.03	0.94	3.10
	SVM	0.96	1.48	0.94	1.87	0.92	2.77	0.95	2.36	0.94	3.16	0.99	1.29	0.97	2.54
IVTD	GP	0.86	5.09	0.81	5.84	0.65	6.16	0.91	4.41	0.82	7.93	0.91	4.75	0.42	5.40
	PLS	0.94	3.28	0.85	5.28	0.81	5.00	0.90	5.53	0.88	5.19	0.98	2.21	0.69	4.50
	SVM	0.91	3.98	0.86	4.98	0.77	5.12	0.92	4.74	0.86	5.60	0.97	2.77	0.65	4.29

GP, Gaussian processes; PLS; partial least square; SVM, support vector machine; R^2^*_c_*, determination coefficient in calibration; RMSE*_c_*, root mean square error in calibration; R^2^*_cv_*, determination coefficient in cross-validation; RMSE*_cv_*, root mean square error in cross-validation; R^2^*_v_*, determination coefficient in external validation; RMSE*_v_*, root mean square error in external validation.

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
