# Peer review of "Predicting Forage Quality of Warm-Season Legumes by Near Infrared Spectroscopy Coupled with Machine Learning Techniques"

_sensors, 2020, doi:10.3390/s20030867_

Reviewer 1 Report

The paper fails to make a clear hypothesis and works on a hypothesis that has already been explored and established The arguments and data presented are not complete in establishing the core idea presented Try to convey your knowledge in simple words. Plan all the different sections that need to be written and start writing in your own words.

Author Response

The paper fails to make a clear hypothesis and works on a hypothesis that has already been explored and established The arguments and data presented are not complete in establishing the core idea presented Try to convey your knowledge in simple words. Plan all the different sections that need to be written and start writing in your own words.

This is a technique-oriented work. It is not possible to develop a standard hypothesis in this case. We disagree that the work is not novel. There are numerous warm-season legumes receiving attention as a forage and thus it is required to develop a low cost, rapid, efficient method to quantify their forage quality. Except some limited information on soybean, there is no accounts reporting the use of near-infrared spectroscopy technique in warm-season legumes. So, it is the first study involving an extensive evaluation of the technique in the forage quality estimations of several warm-season legumes in combination with machine learning methods. Also, We have added text in the introduction for clarity.

 Regarding data presentation and organization of idea/words, we would like the reviewer to be more specific. We do not understand where is the problem.

Round 2

Reviewer 1 Report

The authors have shown efforts to improve the manuscript and this should be well appreciated.

Author Response

Thank you very much for your feedback. We are glad that our work can represent a good contribution to the journal.

Reviewer 3 Report

The improvement is reasonably good. It would be better to discuss uncertainty handling and feature selection with reference e.g. 

Mengbao Fan, etc. Uncertainty metric in model-based eddy current inversion using the adaptive Monte Carlo method, Measurement, Volume 137, 2019,Pages 323-331, https://doi.org/10.1016/j.measurement.2019.01.004.

Author Response

We appreciate this feedback, thanks for the offered corrections and suggestions that have improved the quality of our article.

Reviewer 2: Anonymous

Reviewer 2 Report

The authors have developed a methodology for predicting forage quality of warm-season legumes by near infrared spectroscopy. Although the use of this technology is new to this type of matter, it does not represent a significant advance according to the scope of this journal. In my opinion, and regarding the approach of the experiments, it would have been more interesting to publish them in a journal closer to Food Science or Chemometrics and Statistics. In any case, it is the Editor’s decission to judge the acceptability of this article in "Sensors".

Once stated this, the paper is well explained and could be useful for researchers who wish to compare different sets of machine learning techniques. However, there is one question and one suggestion I would like to make that I think could improve the document:

Lines 128-130. The authors state that they have separated the samples into different parts, however, it is not sufficiently clear whether the models work with all parts together. If a rapid methodology for quality assessment is to be developed and a separation is necessary, this makes the analysis considerably more tedious. Tables 2-6. The numerical data are shown in these tables and further commented on the Results section. It would be very instructive to replace these tables with graphs in which it would be much easier to comprehensively assess the scope of each of the methods developed.

Author Response

The authors have developed a methodology for predicting forage quality of warm-season legumes by near infrared spectroscopy. Although the use of this technology is new to this type of matter, it does not represent a significant advance according to the scope of this journal. In my opinion, and regarding the approach of the experiments, it would have been more interesting to publish them in a journal closer to Food Science or Chemometrics and Statistics. In any case, it is the Editor’s decission to judge the acceptability of this article in "Sensors".

Once stated this, the paper is well explained and could be useful for researchers who wish to compare different sets of machine learning techniques. However, there is one question and one suggestion I would like to make that I think could improve the document:

We appreciate this feedback, thanks for the offered suggestions that improved the quality of our article.

Lines 128-130. The authors state that they have separated the samples into different parts, however, it is not sufficiently clear whether the models work with all parts together. If a rapid methodology for quality assessment is to be developed and a separation is necessary, this makes the analysis considerably more tedious.

Random samples used in the calibration as well as validation procedures were derived from stems, leaves, pods and whole plants. That was a mistake in the text, which has been corrected (Line 111-116). 

Tables 2-6. The numerical data are shown in these tables and further commented on the Results section. It would be very instructive to replace these tables with graphs in which it would be much easier to comprehensively assess the scope of each of the methods developed. 

We agree. Presenting results through graphs is much more instructive than tables. However, in this case, there would be way too many graphs. For instance, if we present just validation results in graphs, there will be a total of 56 graphs in the paper. And three times more if we also count calibration and cross-validation results. Therefore, we preferred tables for the sake of space.

Reviewer 3: Gui Yun Tian

Reviewer 3 Report

The work is interesting. The presentation can be improved as list below.

1. More feature extraction selection and fusion including PCA should be reviewed and discussed with reference including review Near Infrared Spectroscopy and Machine Learning Techniques e.g. 

R Sutthaweekul, etc, Microwave Open-ended Waveguide for Detection and Characterisation of FBHs in Coated GFRP Pipes, Composite Structures, 111080, 2019

Liang Cheng, etc. Impact Damage Detection and Identification Using Eddy Current Pulsed Thermography Through Integration of PCA and ICA, IEEE Sensors Journal 14(5):1655-1663 · May 2014 

2. The training and test samples and results should be discussed in line with major contribution;

3. Mind technical contents and comparison.

Author Response

The work is interesting. The presentation can be improved as list below.

We appreciate this feedback, thanks for the offered suggestions.

More feature extraction selection and fusion including PCA should be reviewed and discussed with reference including review Near Infrared Spectroscopy and Machine Learning Techniques e.g. 

R Sutthaweekul, etc, Microwave Open-ended Waveguide for Detection and Characterisation of FBHs in Coated GFRP Pipes, Composite Structures, 111080, 2019

Liang Cheng, etc. Impact Damage Detection and Identification Using Eddy Current Pulsed Thermography Through Integration of PCA and ICA, IEEE Sensors Journal 14(5):1655-1663 · May 2014 

Thanks, we do not think there is need to discuss feature extraction selection such as PCA, as that was not within the scope of the manuscript.

The training and test samples and results should be discussed in line with major contribution;

Would you please be more specific in what part of the manuscript had such issues?

Mind technical contents and comparison.

We will appreciate if you can provide sections or lines causing trouble to understand.

Editor: Carlos Poblete-Echeverria
Received: 20 December 2019 / Accepted: 04 February 2020